# A Lab-on-a-Chip Approach for the Detection of the Quarantine Potato Cyst Nematode *Globodera pallida*

**DOI:** 10.3390/s23020647

**Published:** 2023-01-06

**Authors:** Maria João Camacho, Débora C. Albuquerque, Eugénia de Andrade, Verónica C. Martins, Maria L. Inácio, Manuel Mota, Paulo P. Freitas

**Affiliations:** 1INIAV, I.P.—National Institute for Agriculture and Veterinary Research, 2780-159 Oeiras, Portugal; 2NemaLab, MED–Mediterranean Institute for Agriculture, Environment and Development, Institute for Advanced Studies and Research, University of Évora, 7006-554 Évora, Portugal; 3INESC-MN- Microsystems and Nanotechnologies, 1000-029 Lisbon, Portugal; 4IST—Instituto Superior Técnico, University of Lisbon, 1049-001 Lisbon, Portugal; 5GREEN-IT Bioresources for Sustainability, ITQB NOVA, 2780-157 Oeiras, Portugal; 6INL—International Iberian Nanotechnology Laboratory, 4715-330 Braga, Portugal

**Keywords:** magnetoresistive biochips, asymmetric PCR, PCN

## Abstract

The potato cyst nematode (PCN), *Globodera pallida*, has acquired significant importance throughout Europe due to its widespread prevalence and negative effects on potato production. Thus, rapid and reliable diagnosis of PCN is critical during surveillance programs and for the implementation of control measures. The development of innovative technologies to overcome the limitations of current methodologies in achieving early detection is needed. Lab-on-a-chip devices can swiftly and accurately detect the presence of certain nucleotide sequences with high sensitivity and convert the presence of biological components into an understandable electrical signal by combining biosensors with microfluidics-based biochemical analysis. In this study, a specific DNA-probe sequence and PCR primers were designed to be used in a magnetoresistive biosensing platform to amplify the internal transcribed spacer region of the ribosomal DNA of *G. pallida*. Magnetic nanoparticles were used as the labelling agents of asymmetric PCR product through biotin–streptavidin interaction. Upon target hybridization to sensor immobilized oligo probes, the fringe field created by the magnetic nanoparticles produces a variation in the sensor’s electrical resistance. The detection signal corresponds to the concentration of target molecules present in the sample. The results demonstrate the suitability of the magnetic biosensor to detect PCR target product and the specificity of the probe, which consistently distinguishes *G. pallida* (DV/V > 1%) from other cyst nematodes (DV/V < 1%), even when DNA mixtures were tested at different concentrations. This shows the magnetic biosensor’s potential as a bioanalytical device for field applications and border phytosanitary inspections.

## 1. Introduction

*Globodera rostochiensis* [1,2] and *Globodera pallida* [3], known as potato cyst nematodes (PCN), are one of the greatest threats to potato crops. These plant parasitic nematodes are originated from the Andes region in southern Peru and spread worldwide due to human activities [4] and lack of phytosanitary measures as they exist nowadays. In Europe, PCN were introduced in the 16–17th century, by means of infested potato tubers, and were also reported throughout North and South America, parts of Asia, Africa and Oceania where potatoes are grown [5]. The golden PCN, *G. rostochiensis*, and the pale PCN, *G. pallida*, are worm-like microscopic endoparasites which feed on potato roots, deteriorating the quality of tubers and reducing their commercial value. In addition, PCN may facilitate the infection of potatoes by opportunistic pathogens, like bacteria and fungi [6], significantly reducing yield, increasing the overall costs of production and imposing trade restrictions. Therefore, upon PCN detection, crop fields are subjected to severe quarantine in many countries, where these nematodes are considered harmful quarantine organisms [7].

Owing to their commercial and environmental impacts, it is essential to detect these species early. A promising tool relies on the use of diagnostic devices in order to implement strategies for an effective integrated pest management. As the morphological identification of these *Globodera* species may be uncertain due to the overlapping morphometric values between both species, molecular confirmation is recommended [8].

PCN molecular identification, described in the European and Mediterranean Plant Protection Organization (EPPO) protocol PM 7/40 [7], is performed through duplex conventional and/or real-time PCR based on the nuclear large subunit ribosomal DNA (rDNA) sequences. Despite the high sensitivity and specificity of these diagnostic methods, the procedures require highly trained staff, are time consuming, the laboratory equipment is expensive and cannot be used in agricultural fields due to the lack of portable devices.

Therefore, due to PCR-based protocol constraints, other methods should be developed, aiming at less practical and technical expertise and at the use of new portable and affordable technological devices for in-field analyses. As a result, different prototypes have been developed concerning the miniaturization of biomolecular methodologies. Microfluidic systems have been used for the automation of experiments and minimization of user intervention [9,10], allowing completely integrated systems, including all steps from sample preparation until DNA amplification [11].

Biosensors, in combination with microfluidics-based biochemical analysis, in a miniaturized device, can rapidly detect the presence of specific nucleotide sequences with high sensitivity and convert the presence of biological compounds into an easy-to-read electrical signal. The detection of DNA amplicons (fragments amplified either by PCR or isothermal reactions) is based on specific target DNA sequence hybridization with a complementary immobilized oligo probe, that can be spotted on chip surfaces in a microarray format [10].

An existing portable electronic reader and magnetoresistive (MR) biochips developed in a collaboration between INESC MN and INESC ID (Lisbon, Portugal) [12,13] were used to discriminate the internal transcribed spacer (ITS) region rDNA of *Globodera pallida* (tested as a model organism) from other related species.

The MR biochip is comprised of an array of 30 spin-valve (SV) sensors which offer particular advantages in terms of reduced size, low limit of detection, analytical sensitivity, high signal-to-noise ratio and integration capability [14,15]. The target molecules are marked with magnetic nanoparticles (MNPs) through biotin–streptavidin interaction, generating a fringe magnetic field when an external magnetic field is applied, proportionally changing the electrical resistance of the sensors [16,17]. Asymmetric PCR products, after amplification, go through a microfluidic system to the probes immobilized on the sensors, allowing their hybridization. The probe sequence can be manually or robotically spotted over the sensing sites and when complimentary target amplicons specifically hybridize, a signal is generated in the transducer [11].

Oligonucleotide probe and specific PCR primers were designed at GMO and molecular biology lab of INIAV (Oeiras, Portugal) to specifically target *G. pallida* based on the ITS-rDNA while avoiding the detection of *G. rostochiensis*, *G. tabacum* and *Heterodera* sp., which can be found in the same fields as *G. pallida*. The analytical specificity and sensitivity of this system was evaluated using detection assays with target DNA amplified by asymmetric PCR using one pair of specific primers and various ratios of template DNA in mixtures of the closely related non-target species *G. rostochiensis*.

Biosensors are growing at a fast pace in human diagnostics, while applications for agriculture remain limited. This work intends to demonstrate the applicability with economic viability of the use of biosensors in agricultural fields for soil pest management or at border phytosanitary inspections facilities.

## 2. Materials and Methods

### 2.1. Sensors

MR sensor microfabrication is described in the work of Martins et al. (2009) [12]. Briefly, the biochip consists of an array of 30 SV sensors passivated with an oxide layer, arranged in six sensing regions, each one containing five active sensors covered with a gold layer, and surrounded by a gold frame for the discrete spotting of the probes. The SV stack consists of the following magnetic thin layers: Ta 2.0 nm/NiFe 2.5 nm/CoFe 2.8 nm/Cu 2.6 nm/CoFe 2.4 nm/MnIr 7.0 nm/Ta 5.0 nm. The sensors are arranged in series of two SVs (active area of 80 × 2.6 µm^2^) electrically contacted by aluminum leads. The sensors’ magnetic response was characterized, obtaining an average MR of 6.0% and sensitivity of 1.3%/mT.

### 2.2. Biochemical Reagents

TE buffer was supplemented with KH_2_PO_4_ (0.1 mM), Tris (10 mM), EDTA (1 mM) and pH was adjusted using HCl (1 M) to 7.4. Phosphate buffer (PB) was prepared from stock solutions of Na_2_HPO_4_ and NaH_2_PO_4_ at 0.2 M and pH 7.2. PB-Tween20 consisted on PB buffer with 0.02% (*v*/*v*) of Tween^®^ 20 from Promega (Madison, WI, USA). All solutions were prepared with ultra-pure grade water.

The customized primers and probes were synthesized by Eurogentec (Seraing, Belgium).

The Magnetic Nanoparticles were nanomag^®^-D from Micromod (Rostock, Germany), with a diameter of 250 nm and 75–80% (*w*/*w*) magnetite in a matrix of dextran (40 kDa), and streptavidin coated. The particles had a magnetic moment of ~1.6 × 10^−^^16^ Am^2^ for a 1.2 kA/m magnetizing field and a susceptibility of χ~4.

### 2.3. Nematode Samples

Nematode isolates of *G. pallida*, *G. rostochiensis*, *G. tabacum*, *Heterodera* sp. and different mixtures of *G. pallida*/*G. rostochiensis* (Table 1) were obtained at the Nematology lab of INIAV (NemaINIAV, Oeiras, Portugal).

The extraction of total DNA was always conducted using the Qiagen DNeasy Blood and Tissue kit (Life Technologies, Carlsbad, CA, USA), following the manufacturer’s instructions. Genomic DNA was quantified using the thermo-NANODROP 2000 spectrophotometer (Thermo Fisher Scientific, Waltham, MA, USA) and stored at −20 °C until further use. DNA extracts were used directly for the PCR reactions without any additional purification step. Total DNA extraction, purification and conservation was performed as described in Camacho et al. [18].

### 2.4. Globodera pallida Probe and Primer Sequence Design

The nucleotide sequences of the “3′end18S-ITS1-5.8S-ITS2-5′end28S” rDNA region used to design the probe specific for the detection of *G. pallida* were acquired during a previous study to develop a new LAMP assay [18,19]. The primers B2 and F3 were selected as the forward and the reverse primers, respectively, to amplify a 141 bp biotinylated product. B2 was biotinylated on the 5′ end. The detection of this product, by immobilization, needs a probe which was labelled with a thiol group and a 15-mer poli-T sequence at the 5′ end for immobilization purposes.

Primers and probe’s sequences and characteristics are summarized in Table 2. Additionally, a probe sequence not related with any target sequence was used as the negative control. The primer properties (were indicated by the manufacturer—Eurogentec (Seraing, Belgium). The probe properties (Table 2), including guanine and cytosine (GC) content, melting temperature (Tm) and change in free energy of hybridization (∆G), were calculated using the IDT Oligo Analyzer tool.

### 2.5. Asymmetric PCR Amplification

The ITS-rDNA was amplified by an asymmetric PCR, with a primer ratio of 10:1 (Fw:Rv). PCR reactions were performed in a 25 µL final volume containing 5 µL template DNA, 5 µL GoTaq Flexi PCR buffer (2×), 5 µL MgCl_2_ (25 mM), 0.4 µL dNTPs (10 mM), 0.5 µL GoTaq Flexi DNA Polymerase (Promega, Madison, USA), 0.375 µL of F3 primer (10 µM), 3.75 µL of b-B2 primer (10 µM) and 4.975 µL of DNA-free water. The amplification profile for ITS-rDNA consisted of an initial denaturation of 94 °C for 2 min followed by 35 cycles at 94 °C for 30 s, 55 °C for 30 s, and 72 °C for 15 s and a final extension of 72 °C for 7 min. The amplified products were visualized using the VersaDoc Gel Imaging System (Bio-Rad, Hercules, CA, USA) after being electrophoresed at 5 V/cm in 0.5× SGTB buffer (GRISP, Porto, Portugal) and in a 1.5% agarose gel stained with ethidium bromide (0.5 µg.mL)^−1^. Possible contaminations were checked by including negative controls (no template control—NTC) in all amplifications.

### 2.6. Detection Assays in the Biochip Platform

Prior to probe immobilization, biochips underwent a cleaning procedure described in Viveiros et al. (2020) [20].

The probes designed for *G. pallida* detection were diluted in the TRIS-EDTA buffer to a concentration of 5 µM and immobilized by manual spotting on the biochip surface (Figure 1—Probe immobilization). Each spot consisted of a drop volume of 1 µL. After spotting, the probes were left to immobilize for 1 h in a humid chamber at room temperature.

The biochip platform was fabricated by INESC ID and INESC MN (Lisbon, Portugal) as described by Germano et al. (2009) [13]. The sensor functionalized with the *G. pallida* probe was inserted in the platform and an U-shaped PDMS microfluidic system was placed over the sensor to transport the reagents, in sequential order, over the sensing area (Figure 1) [21], All reagents were loaded at a flow rate of 50 µL/min, with the help of a syringe pump (NE-300, NEW ERA, Buffalo, NY, USA). First, sensors were washed with PB buffer to remove unbound probes, followed by the loading of 10 µL of target asymmetric PCR product to cover the sensing sites (Figure 1—Hybridization). The hybridization was left to occur for 30 min, after which, unbound target molecules were washed off with PB buffer. Next, the MR measurement was initiated by first acquiring the baseline voltage of the sensors for 5 min, followed by the injection of the MNPs (10× diluted from stock) into the PDMS channel which were then left to incubate over the sensors for 20 min (Figure 1). After the resistance signal of the sensors saturated, the unbound particles were washed off for 5 min at continuous flow, or until signal stabilization. In total, data acquisition took about 30 min. The main steps of the measurement are represented in Figure 1.

The sensors were biased with 1 mA DC current, and the MNPs magnetized with an external AC magnetic field of 13.5 Oe at 211 Hz and a DC field of 35 Oe. A voltage signal was acquired for each sensor and the data was recorded (Figure 2).

### 2.7. Data Analysis

The binding signals are differential voltage values identified as ∆V_ac_^binding^, calculated from the difference between the sensor baseline (V_ac_^sensor^) and the signal originating from the specifically bound MNPs over the sensor (V_ac_^particles^). The ∆V_ac_^binding^ signal is then normalized by the sensors’ baseline and taken as the final output read-out signal (∆V_ac_^binding^/V_ac_^sensor^). Additionally, in each substrate, a reference spot with an unspecific probe (whose target is *Chikungunya*—Table 2) was performed to remove the influence of unspecific binding. The measurement curves on Figure 2 correspond to the sensors used to detect (I) target DNA and (II) a non-complementary target.

## 3. Results and Discussion

### 3.1. Asymmetric PCR

DNA samples of *Globodera pallida*, *G. tabacum*, *G. rostochiensis* and *Heterodera* sp. were amplified by asymmetric PCR using the pair of primers indicated in Table 2, designed on a region of the ITS-rDNA conserved among different isolates of *G. pallida* and variable among other species. Figure 3 presents the agarose gel of asymmetric PCR amplification products.

For all targets, more than one band was observed. These bands correspond to both double strand (dsDNA) and single strand DNA (ssDNA) products from the asymmetric PCR. The limiting primer was involved in the production of dsDNA since the first reaction cycle until it was fully consumed, when the ssDNA production started, supported by the forward primer in excess.

### 3.2. Detection Assays in the Biochip Platform

Detection assays were performed in the magnetoresistive biochip device with the target amplified by asymmetric PCR of genomic DNA samples. The data acquired from each sensor was analyzed as previously described. Different samples were tested against the specific probe for *G. pallida* and a negative control probe was used as a reference signal. At least three replicated measurements were performed for each sample, corresponding to the detection signal of at least 12 sensors in each measurement. The results obtained are summarized in Figure 4. Each bar of the graphic represents the normalized signal acquired from the probe against the *G. pallida*, mixed samples and non-target species PCR products. The threshold value (dashed line) was obtained from the value between the highest non-specific signal achieved against a non-complementary target and the lower specific signal obtained against a complementary target (standard deviation was taken into consideration). Above the threshold value, the detection signal was considered positive.

The tested probe showed specific signals against its complementary target (*G. pallida*) without significant cross reactivity, even when using pooled samples with *G. pallida* mixed with *G. rostochiensis* (ratios of 1/5, 1/19 and 1/40), corresponding to a diagnostic sensitivity of one (1) juvenile. All samples with *G. pallida* DNA obtained detection signals higher than 1% (1.9 ± 0.77%) and all samples with non-target DNA obtained detection signals lower than 1% (−0.04 ± 0.44%). *Globodera rostochiensis* samples, as expected due to PCR product amplification be the closest related to *G. pallida*, obtained higher detection signals than the others non-target species (*G. tabacum* and *Heterodera* sp.), even so lower than 1%. These data are in line with previous studies, whose reports show a positive detection signal of 1.8 ± 0.7% and a negative control of 0.4 ± 0.3% [20].

The results demonstrate the specificity of the probe which reliably discriminates *G. pallida* from other cyst nematodes. The MR biosensor shows specific signals for qualitative *G. pallida* detection through a double specific control—PCR and probe hybridization efficiency—avoiding false positives for non-targets samples, such as *G. rostochiensis*, *G. tabacum* and *Heterodera* sp. This approach shows great promise for field application in the early detection and surveillance of plant soil pests and in assisting the implementation of management practices to reduce the risk of infestations. Another possible application is at border phytosanitary inspections. New technologies are in high demand in the agricultural market to address the problem of plant pest detection and there is a clear opportunity for new developments in portable devices for agriculture applications. Further improvement of this technique will include an isothermal amplification of DNA (e.g., LAMP—Loop Isothermal AMPlification) [22] to avoid the need for high temperatures which is the major impediment for its application in-field, and the use of Flinders Technology Associates (FTA) card protocol for DNA extraction on-site [23].

Despite not being the goal of this work, whose purpose was to qualitatively detect *G. pallida* (tested as a model organism), other works have achieved the simultaneous multiplex detection of different pathogens based on an asymmetric PCR protocol coupled with a magnetic array biochip functionalized with species-specific probes [20,24]. In the future, a multiplex detection protocol can be designed for the detection of different *Globodera* species using a single pair of primers in asymmetric PCR to indiscriminately amplify any target *Globodera* sp. in conjunction with species-specific sensor-immobilized oligonucleotide probes.

## 4. Conclusions

Recently, we have seen an increasing need for new detection methods, mainly for plant pests and diseases.

An essential consideration in phytosanitary study is the cost–benefit ratio. Although the use of biosensors in human diagnosis is expanding quickly, there are still few applications in agriculture. With this work, we tried to manage plant pests in agricultural fields by integrating the use of biosensors. This activity is in line with the European Green Deal, which acknowledges digitization as a tool to enhance output by lowering the impact of pests and diseases, improving productivity and enabling an ecological transition (reduction in pesticide applications).

Nowadays, *G. pallida* constitutes a big threat to all potato-producing regions. Its management is being affected by the few attractive *G. pallida* resistant/tolerant potato cultivars, compared to several cultivars with a high tolerance to *G. rostochiensis*, which is leading to *G. pallida* selection. Therefore, for field detection, we used a magnetoresistive biochip device for the specific identification of *G. pallida* by targeting the ITS-rDNA sequence. The primers designed for the PCR amplification in combination with the probe specifically detected *G. pallida* in DNA extracts. No false positives were observed with other closely related species. These observations show that the tested biosensors are highly specific for detecting *G. pallida* even in samples infested with cysts of other *Globodera* species.

It is possible to investigate this technology for the detection of other organisms and plant pests and pathogens. It does not require specific knowledge or experience from the operator. Thus, this method can be considered very beneficial for the surveillance and disease plant control purposes.

## Figures and Tables

**Figure 1 sensors-23-00647-f001:**
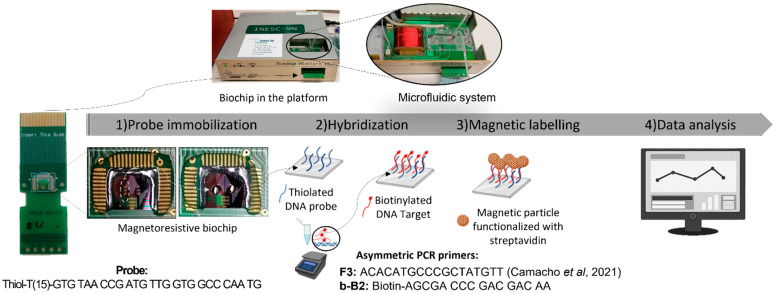
Schematic representation of the main steps involved in a measurement [19].

**Figure 2 sensors-23-00647-f002:**
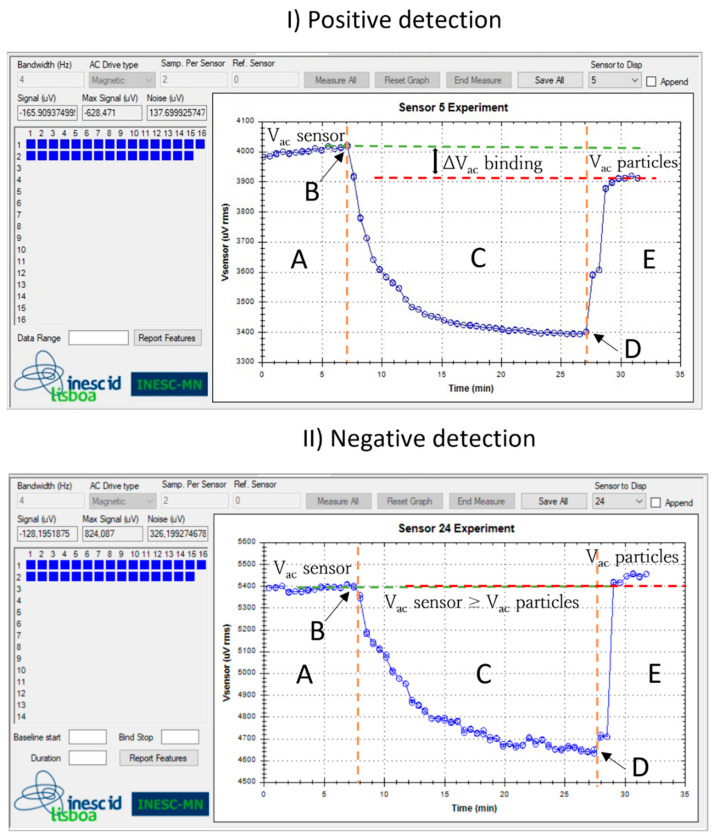
Voltage signal acquired from two sensors. Both measurements occur through five phases: (A) acquisition of the sensor baseline signal (V_ac_^sensor^); (B) magnetic particle addition; (C) decreasing signal due to the magnetic particles settling down over the sensor; (D) saturation signal; and (E) washing step and final signal corresponding to the presence of target bound magnetic particles over the sensor (V_ac_ ^particles^): (**I**) positive detection event: hybridization with a complementary target DNA (*Globodera pallida*) labeled with 250 nm magnetic particles ending at a lower voltage and (**II**) negative detection event: non-hybridization with a non-target DNA (*G. rostochiensis*) ending at an equal voltage value.

**Figure 3 sensors-23-00647-f003:**
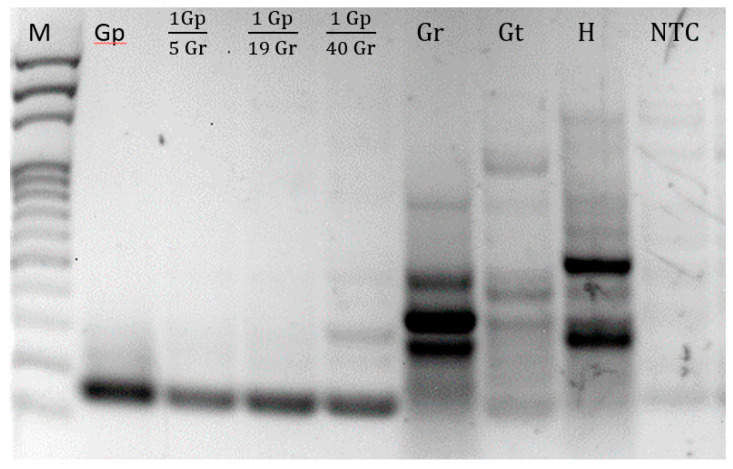
Agarose gel of the amplified products obtained with asymmetric PCR using F3 and b-B2 primers. M = 100 bp DNA Ladder (Thermo Scientific); Gp = *Globodera pallida*.; Gp/Gr = ratios of *G. pallida* and *G. rostochiensis*; Gr = *G. rostochiensis*; Gt = *G. tabacum*; H = *Heterodera* sp.; NTC = negative control (no DNA template).

**Figure 4 sensors-23-00647-f004:**
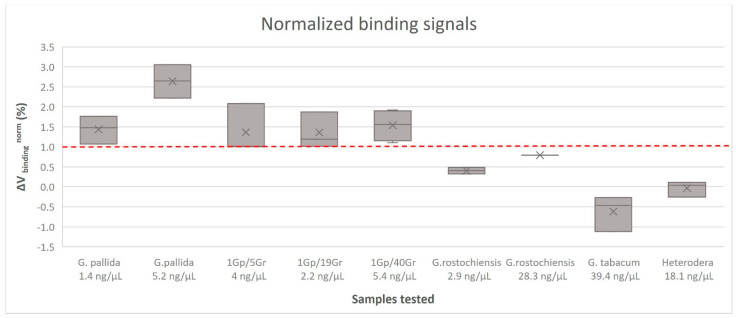
Normalized binding signals obtained for each sample group, obtained from asymmetric PCR against the specific probe for *Globodera pallida* detection. The error bars are standard deviations coming from at least 12 sensors acquired from three measures for each sample. The dashed line represents the threshold, a minimum value above which a detection signal is considered positive. DNA of pure samples (*Globodera pallida*, *G. rostochiensis*, *G. tabacum* and *Heterodera* sp.) was extracted from cysts and DNA extracted from pooled samples was extracted from mixed juveniles.

**Table 1 sensors-23-00647-t001:** Samples from Portugal and The Netherlands used for on-chip assays.

Species	Isolate	Origin	ng/µL
*G. pallida*	MK791521	Portugal	5.2
*G. pallida*	NPPO-NL Pa3 HLB	The Netherlands	1.4
1 Gp/5 Gr	MK791521/MK791264	Portugal	4
1 Gp/19 Gr	MK791521/MK791264	Portugal	2.2
1 Gp/40 Gr	MK791521/MK791264	Portugal	5.4
*G. rostochiensis*	MK791264	Portugal	28.2
*G. rostochiensis*	NPPO-NL Ro1 HLB	The Netherlands	2.9
*G. tabacum*	NPPO-NL C6876	The Netherlands	39.4
*Heterodera* sp.	SV-18-10003	Portugal	18.1

**Table 2 sensors-23-00647-t002:** Sequence, size, GC content, and melting temperature (Tm) of a universal pair of primers designed based on the ITS-rDNA of *Globodera pallida* and the change in free energy of hybridization (∆G) of the oligonucleotide probe specifically designed to target *Globodera pallida,* and of the negative control probe.

Primers	Sequence (5′-3′)	Size (bp)	GC%	Tm (°C)	∆G (kcal/mol)
F3—Reverse primer (Rv) [19]	ACA CAT GCC CGC TAT GTT	18	50	54	
b-B2—Forward primer (Fw)	Biotin-AG CGA CCC GAC GAC AA	16	62.5	52	
*G. pallida*	Thiol 15T GTG TAA CCG ATG TTG GTG GCC CAA TG	26	53.8	62.1	−51.85
*Chikungunya*	Thiol 15T CGC ATA GCA CCA CGA TTA G	19	52.6	53.4	−36.7

## Data Availability

Not applicable.

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
