# Peer review of "A Lab-on-a-Chip Approach for the Detection of the Quarantine Potato Cyst Nematode Globodera pallida"

_sensors, 2023, doi:10.3390/s23020647_

Round 1
Reviewer 1 Report
The paper describes Globodera pallida detection via developed biosensors in combination with microfluidics. The authors study the cross-selectivity with other cyst nematodes and demonstrate the specificity towards potato cyst nematode of the developed magnetoresistive biosensors. The paper could be accepted after a major revision.
Comments:
1. Please specify the final concentrations you used in Figure 4 for all analytes. Is there any dependence in response to different concentrations of Globodera pallida only in buffer solution? What is the detection limit of your biosensing platform to potato cyst nematode? Could you provide in a supplementary file the response curves for all measured species?
2. I'm not sure that a 1 or 1.5 % difference in response causes "very high specificity for qualitative G. pallida detection". Please provide a comparison with the literature or rephrase the sentence.
3. The Materials and Methods section could be shortened or moved in part to supplementary.
4. In Table 1 no mixtures of G. pallida/G. rostochiensis are presented as said on page 3 lines 132-133.
5. What was the idea to use sample molecules from two different suppliers? Have you tried to compare the response for the same concentration in the buffer?
Reviewer 2 Report
The manuscript titled "A lab-on-a-chip technique for the detection of the quarantine potato cyst worm Globodera pallida" is an interesting study that deserves to be published. Because of Globodera pallida widespread prevalence and detrimental impact on potato production, it has importance throughout the Europe. Hence, a quick and accurate diagnosis of PCN was essential for the implementation of control measures as well as surveillance programmes.
#The authors have done good efforts to overcome the shortcomings of the current approaches in attaining early detection. The lab-on-a-chip devices can swiftly and accurately detect the presence of certain nucleotide sequences with high sensitivity and convert the presence of biological components into an understandable electrical signal by combining biosensors with microfluidics-based biochemical analysis.
#In this study, a unique DNA probe sequence and PCR primers were developed for usage in a magneto resistive biosensing platform to amplify the internal transcribed spacer region of the ribosomal DNA of G. pallida.
# The results demonstrate the suitability of the magnetic biosensor to detect PCR target product and the specificity of the probe, which consistently distinguishes G. pallida (DV/V > 1%) from other cyst nematodes (DV/V 1%), even when DNA mixtures were tested at different concentrations.
#This shows the magnetic biosensor's potential as a bioanalytical device for field applications and border phytosanitary inspections.
#An essential consideration in phytosanitary study is the cost-benefit ratio. Although the use of biosensors in human diagnosis is expanding quickly, there are still few applications in agriculture.
#The authors tried to manage plant pests in agricultural fields by integrating the use of biosensors. This activity is in line with the European Green Deal, which acknowledges digitization as a tool to enhance output by lowering the impact of pests and illnesses, improving productivity, and enabling an ecological transition.
# It is possible to investigate this technology for the detection of other species and plant diseases. As a result, this technique is particularly beneficial for monitoring and controlling disease in plants.
# The study was executed and organised well overall, and with only minimal changes, it might be accepted.
Round 2
Reviewer 1 Report
All the comments were answered by the authors. The manuscript should be published in its current form.